# Plasticity of Expression of Stem Cell and EMT Markers in Breast Cancer Cells in 2D and 3D Culture Depend on the Spatial Parameters of Cell Growth; Mathematical Modeling of Mechanical Stress in Cell Culture in Relation to ECM Stiffness

**DOI:** 10.3390/bioengineering12020147

**Published:** 2025-02-04

**Authors:** Małgorzata Szostakowska-Rodzoś, Mateusz Chmielarczyk, Weronika Zacharska, Anna Fabisiewicz, Agata Kurzyk, Izabella Myśliwy, Zofia Kozaryna, Eligiusz Postek, Ewa A. Grzybowska

**Affiliations:** 1Molecular and Translational Oncology Department, Maria Sklodowska-Curie National Research Institute of Oncology, Roentgena 5, 02-781 Warsaw, Poland; malgorzata.szostakowska-rodzos@nio.gov.pl (M.S.-R.); mateusz.chmielarczyk@nio.gov.pl (M.C.); weronika.juras97@gmail.com (W.Z.); anna.fabisiewicz@nio.gov.pl (A.F.); agata.kurzyk@nio.gov.pl (A.K.); izabella.mysliwy@nio.gov.pl (I.M.);; 2Institute of Fundamental Technological Research, Polish Academy of Sciences, Pawińskiego St. 5B, 02-106 Warsaw, Poland; epostek@ippt.pan.pl

**Keywords:** breast cancer, E-cadherin, mechanical stress, mathematical modeling, 3D bioprinting, complex systems, cell modeling, finite element method

## Abstract

The majority of the current cancer research is based on two-dimensional cell cultures and animal models. These methods have limitations, including different expressions of key factors involved in carcinogenesis and metastasis, depending on culture conditions. Addressing these differences is crucial in obtaining physiologically relevant models. In this manuscript we analyzed the plasticity of the expression of stem cell and epithelial/mesenchymal markers in breast cancer cells, depending on culture conditions. Significant differences in marker expression were observed in different growth models not only between 2D and 3D conditions but also between two different 3D models. Differences observed in the levels of adherent junction protein E-cadherin in two different 3D models suggest that spatial parameters of cell growth and physical stress in the culture may affect the expression of junction proteins. To provide an explanation of this phenomenon on the grounds of mechanobiology, these parameters were analyzed using a mathematical model of the 3D bioprinted cell culture. The finite element mechanical model generated in this study includes an extracellular matrix and a group of regularly placed cells. The single-cell model comprises an idealized cytoskeleton, cortex, cytoplasm, and nucleus. The analysis of the model revealed that the stress generated by external pressure is transferred between the cells, generating specific stress fields, depending on growth conditions. We have analyzed and compared stress fields in two different growth conditions, each corresponding to a different elasticity of extracellular matrix. We have demonstrated that soft matrix conditions produce more stress than a stiff matrix in the single cell as well as in cellular spheroids. The observed differences can explain the plasticity of E-cadherin expression in response to mechanical stress. These results should contribute to a better understanding of the differences between various growth models.

## 1. Introduction

Most of the current approaches to cancer research and preclinical drug testing are based on 2D monolayer cell cultures, but there is a growing understanding that other models are needed to more accurately replicate the actual response of cells within the organism, including the expression and response to drugs [1,2]. 3D cell models were developed to mimic the structural and functional complexity of in vivo tissues. Since the information acquired from 2D monolayer culture is still a cornerstone for the research, it is necessary to establish whether there are differences in the expression of crucial markers between 2D conditions and more advanced 3D models.

Until now, differences in markers’ expression have been observed between 2D and 3D conditions. It was reported that under 3D conditions the expression of stem cell markers is higher [3,4], including research in MCF7 cell line [5]; however, this effect might be caused not by the induction of their expression but by the selection of stem-cell phenotypes during spheroid formation. Epithelial–mesenchymal transition (EMT) markers have been also reported to change between 2D and 3D conditions [3,6,7]. These differences can be crucial for studies on cancer progression and metastasis, because the expression of these markers is critical in such research.

Despite these advances, significant gaps remain in understanding how different 3D culture conditions impact the plasticity of epithelial and mesenchymal markers, which are crucial for cancer progression and metastasis. There are different types of 3D cell culture, ranging from floating spheroids that rely on the self-aggregation of cells in scaffold-free conditions to different types of scaffold-based spheroids, including cells cultured in specific hydrogels on standard plates or specialized 3D culture platforms [8,9,10]. Recently, great efforts have been made to manufacture and engineer optimal drug testing platforms, characterized by high efficiency and reproducibility, including different micro-/nanopatterned scaffolds [11], bioreactor systems [12], microcapsulation [13] and magnetic levitation [14]. Another approach is represented by bioprinted 3D models, which are also of interest in the field of tissue engineering and regenerative medicine. These bioprinted cultures, especially multicellular, seem to be the closest approximation of the real tumor [15].

The multitude of 3D models raises the question of whether all these models are equivalent in terms of the expression of some specific markers commonly used in cancer research. Previous studies have predominantly focused on a single 3D model, limiting comparative insights across various 3D environments. Furthermore, the mechanical stresses experienced by cells within these models are often overlooked, despite evidence that mechanical forces influence gene expression and cell morphology [12].

This study aims to address the mentioned gaps by investigating the plasticity of stem cell and EMT markers in breast cancer cells grown in different 3D culture systems. The research integrates experimental analysis with a mathematical model of mechanical stress to evaluate how ECM stiffness affects cellular behavior. By comparing scaffold-free spheroids and scaffold-based 3D bioprinted cultures, this study seeks to uncover the underlying mechanisms driving marker plasticity and their implications for cancer progression.

The results presented here confirm a higher expression of stem-cell markers in 3D vs. 2D conditions, but the expression of the epithelial marker E-cadherin displayed intriguing variability depending on the 3D model. Since there is an obvious difference in the mechanical stress imposed on cells that form free-floating spheroids and cells in a culture on a scaffold, we addressed this difference by creating a mathematical model of a bioprinted culture. This model was used to compare stress fields generated in relation to the stiffness of the ECM (different Young’s modulus, mimicking different 3D systems), to substantiate our interpretation that differences in mechanical stress in cell–cell junctions imposed by growth conditions translate into an observed high plasticity of E-cadherin expression. These findings may be important for the proper interpretation of the results obtained using different 3D models of cancer cells, for practical drug-screening purposes as well as for the basic research on the significance of E-cadherin expression for the metastatic process.

## 2. Materials and Methods

### 2.1. Cell Lines

MCF7 (ATCC, American Type Cell Culture, Manassas, VA, USA), T47D (DSMZ, Deutsche Sammlung von Mikroorganismen und Zellkulturen, Leibniz Institute, Braunschweig, Germany) and MDA-MB-231 (ATCC) cell lines were cultured in DMEM supplemented with 10% fetal bovine serum (Thermo Fisher Scientific, Waltham, MA, USA) or for 3D culture conditions in 3dGRO Basal Medium (Sigma Aldrich, Saint Louis, MO, USA). All cell lines were authenticated by Eurofins Genomics (Ebersberg, Germany).

### 2.2. Spheroid Culture

Cell suspensions were prepared by filtering trypsin-detached cells through a 40 μm Cell Strainer (Sigma Aldrich), and 8 × 104 cells were seeded into ultra-low attachment (ULA) X-well round-bottomed plates (Corning, NY, USA) in 2 mL of 3dGRO Basal Medium (Sigma Aldrich) and cultured for 7 or 14 days, with medium supplementation every 2 days.

### 2.3. Matrigel Culture

Liquid Matrigel was defrosted on ice, pipetted into a six-well plate (616 μL per well) and incubated for 30 min to solidify. Cells were filtered through a Nylon Blue Cell Strainer, pore size 40 μm (Sigma Aldrich), to disperse clumps and seeded at a density of 48,000 per well in 2 mL 3dGRO Basal Medium (Sigma Aldrich) containing 2% Matrigel. Cells were cultured for 1 to 2 weeks, and medium was replaced every 48–72 h. Cells were collected by trypsinization (30 min), washed with cold PBS and analyzed.

### 2.4. Immunofluorescence

Cells from the 3D spheroid culture were harvested on the seventh day of the culture for staining, and 2D cultured cells were harvested at the same time-points at the confluency ~80%. Cells were fixed using 4% PFA (paraformaldehyde) for 15 min. The permeabilization was achieved with permeabilization buffer: 1×PBS, 1%BSA, 0.2% Triton-X for 20 min. Cells were incubated with primary antibodies overnight and washed five times with 1×PBS, and secondary antibodies were added. The staining was carried out at 4 °C for 2 h. The cells were washed five times with 1×PBS, suspended in ddH_2_O, smeared on the microscopic glass and mounted. Imaging was accomplished using a Zeiss Axio observer Z1 LSM 800 confocal microscope (Carl Zeiss AG, Oberkochen, Germany). Primary antibodies: E-cadherin, Vimentin (Cell Signaling, Danvers, MA, USA), SOX2-DyLight550, OCT-4-AlexaFluor488 (Thermo Fisher), secondary antibodies: Anti-Mouse IgG-AlexaFluor488 (Thermo Fisher), Anti-Rabbit IgG-HRP (Abcam, Cambridge, UK), dyes: DAPI (4′,6-Diamidino-2-Phenylindole, Dihydrochloride) (Thermo Fisher), calcein, Hoechst 33342 (Sigma Aldrich).

### 2.5. qPCR

Quantitative PCR was performed as described [16]. Briefly, MCF7 and T47D cells were harvested under the indicated conditions and used for total RNA preparation (PureLink RNA mini kit; Thermo Fisher Scientific), followed by treatment with recombinant DNase I (Roche, Basel, Switzerland). 1 μg of the obtained RNA was used for cDNA synthesis using Superscript III (Thermo Fisher Scientific). cDNA was quantified by quantitative PCR on an ABI Prism 7500 real-time PCR system using TaqMan Gene Expression Assays (Thermo Fisher Scientific, SOX2, Hs04234836_s1, POU5F1, Hs00999632_g1, GAPDH, Hs02786624_g1). The reaction was carried out under the following cycling conditions: initial step of 10 min at 95 °C, followed by 45 cycles of a 15 s denaturation step at 95 °C and then 60 s of annealing and extension at 60 °C. The ΔΔCT method was used for calculating mRNA expression levels.

### 2.6. Western Blot

Cells were harvested and lysed with RIPA (150 mM NaCl; 1% NP.-40; 50 mM Tris pH 8.0; 0.1% SDS; 0.5% sodium deoxycholate) buffer with proteinase inhibitors. Protein concentration was measured using Bradford Assay Reagent (Thermo Fisher). 20 µg of protein extract was boiled with 5× loading buffer, and the proteins were separated on 12% SDS-PAGE and transferred to the PVDF membrane (Merck, Darmstadt, Germany), previously activated with methanol. The membrane was blocked with the blocking buffer (5%-milk in 1×PBS) for 1 h. Primary antibodies were incubated overnight. The membrane was washed five times in TBS-T buffer, incubated with secondary antibodies for 2 h and then washed again five times with TBS-T buffer. Final imaging was carried out with the use of a Mini HD 4 UVITEC system.

### 2.7. Biofabrication of 3D Breast Cancer Models

The architecture of the bioprinted 3D breast cancer models was designed using the SLICER 4.0 3D software. Each 3D model was created with dimensions of 5 × 5 × 1 mm. The models were printed using a BIOX bioprinter (Cellink, Göteborg, Sweden) and cultured in vitro. Based on our previous research [4], the BIOINK hydrogel (Cellink) based on alginate was selected due to its stability, high biocompatibility and suitability for long-term 3D cell cultures. The 3D-MCF-7 model consisted of three layers of the MCF-7 cell line, while the 3D-MDA model also comprised three layers of the MDA cell line. Both cell lines were used at passage 7. Printing was conducted in 24-well plates (Thermo Fisher, Costar) at a temperature of 22–25 °C, using a 22 G nozzle with a thickness of 0.40 mm and 60% rectilinear infill. The pressure ranged from 18 to 20 kPa. Cell suspensions were mixed with BIOINK hydrogel (Cellink) in a 1:1 ratio and extruded into the desired patterns. All 3D constructs were stabilized by crosslinking with a 50 mM CaCl_2_ solution for 5–10 min. After crosslinking, the DMEM medium was added, and the constructs were cultured under standard conditions. The next day, the medium was replaced with fresh DMEM, and the constructs were incubated under standard culture conditions. Bioprinted constructs were cultured for 8 weeks. 3D models were evaluated morphologically, and five out of 34 were selected for imaging. Live cells were stained by incubation with calcein (Thermo Fisher Scientific, working solution: 10 μM 2 h) and Hoechst 33342 (Thermo Fisher Scientific, 1:1000, 10 min). The imaging was performed using a Zeiss LSM800 Axio Observer.Z1/7 confocal microscope and objective EC Plan Neofluar 10×/0.3 M27 (for live images). Images of 54, 199, 241, 279 and 253 focal planes of the spheroid culture were generated and evaluated for modeling. The image used for modeling contained 279 focal planes (Z-stacks) over the distance of 417 μm.

### 2.8. Mathematical Model

#### 2.8.1. Methods

Cells and the extracellular matrix (ECM) were modeled using a finite element method [17]. The modeling was carried out with the Abaqus program https://discover.3ds.com (accessed on 12 December 2024). The geometrical model was prepared using the GiD program https://www.gidsimulation.com (accessed on 12 December 2024). The role in the modeling is presented in Section 2.8.2. The general purpose finite element system is the one of the most often used in industry. The program has all the attributes to model the presented cell–matrix system: it possesses the necessary elements like solid four-node tetrahedra, triangular membranes, and prestressed bars. The elements can be assembled in one finite element system. The solver of the system controls the solution regarding the shape of the elements at the preprocessing stage. During the solution, the system of equation correctness is controlled. It is accomplished during the Newton–Raphson iteration process. The solver is parallelized. Concerning the presented models, the solution takes about 1 h using 48 cpus. In the analysis, the nonlinear geometry is taken into account [18,19]. This is due to the presence of the prestressing forces in the cytoskeleton [20,21]. Therefore, the nonlinear part of the strain tensor is included [22]. Since the incremental solution is used in the Abaqus 2024 Standard program, the strain increment reads:(1)ΔE=Δe+Δη,
where Δ**e** and Δ**η** are the linear and nonlinear parts of the strain increment, respectively. They are of the form:(2)$$\;\Delta e=A\;\Delta u,\;\;\;\;\;\;\;\;\Delta \eta =\frac{1}{2}\;A_{L}\left(\Delta u’\right)\Delta u^{\prime },$$
where Δu and ∆u’ are the displacement’s increment vector and vector of the increment of the displacement derivatives with respect to Cartesian coordinates, respectively. The displacement vector components are (u, v, w). The symbols **A** and **A**_L_ stand for the linear and nonlinear operators, respectively, as follows:(3) A=∂∂x000∂∂y000∂∂z   ∂∂y∂∂z0∂∂x0∂∂z0∂∂x∂∂yT(4)$$A_{L}=\left[\begin{array}{ccccccccc} \Delta u{,}_{x} & 0 & 0 & \Delta v{,}_{x} & 0 & 0 & \Delta w{,}_{x} & 0 & 0\\ 0 & \Delta u{,}_{y} & 0 & 0 & \Delta v{,}_{y} & 0 & 0 & \Delta w{,}_{y} & 0\\ 0 & 0 & \Delta u{,}_{z} & 0 & 0 & \Delta v{,}_{z} & 0 & 0 & \Delta w{,}_{z}\\ \Delta u{,}_{y} & \Delta u{,}_{x} & 0 & \Delta v{,}_{y} & \Delta v{,}_{x} & 0 & \Delta w{,}_{y} & \Delta w{,}_{x} & 0\\ 0 & \Delta u{,}_{z} & \Delta u{,}_{y} & 0 & \Delta v{,}_{z} & \Delta v{,}_{y} & 0 & \Delta w{,}_{z} & \Delta w{,}_{y}\\ \Delta u{,}_{z} & 0 & \Delta u{,}_{x} & \Delta v{,}_{x} & \Delta v{,}_{z} & 0 & \Delta w{,}_{z} & 0 & \Delta w{,}_{x} \end{array}\right]$$

The results of the analysis are the displacement and stress fields. In the results section, the displacement fields are shown. In the case of the cytoskeleton, the uniaxial stress is presented. Then, the Huber–Mises–Hencky (HMH) stress is given showing an effort of the material. The HMH stress reads as follows:(5)$$\;\sigma_{HMH}=\sqrt{\frac{1}{2}\left[{\left(\sigma_{x}-\sigma_{y}\right)}^{2}+{\left(\sigma_{y\;}-\sigma_{z}\right)}^{2}+{\left(\sigma_{z}-\sigma_{x}\right)}^{2}\right]+3\left(\tau_{xy}^{2}+\tau_{yz}^{2}+\tau_{zx}^{2}\right)}$$
where $\left\{\sigma_{x},\;\sigma_{y},\sigma_{z},\tau_{xy},\tau_{yz},\tau_{zx}\right\}$ are the elements of the stress tensor that describe the 3D state of stress.

The solution was performed in the Abaqus program in two steps. The first step concerns the evaluation of the stress state after introducing the prestressing forces in the cytoskeleton. There is no external pressure at this step. The solution shows the state of stress in the system due to prestressing. The second step is applied to impose the external pressure load. It shows the combined state of stress due to the external loading and prestressing of the cytoskeletons. Due to prestressing and the assumption of nonlinear geometry, the solution is performed using the incremental procedure with Newton–Raphson iterations at each loading increment. The prestress step is divided into two increments, and the loading step is divided into six increments. Each increment requires only one iteration.

#### 2.8.2. Geometry

The geometry of the systems is shown in Figure 1. The cells are embedded in ECM. A simplified model of the environment is adopted; ECM is modeled as an elastic, nearly incompressible medium. The ECM for a single cell is discretized with 2,248,103 tetrahedral elements, while the ECM for the group of 18 cells is discretized with 2,070,659 tetrahedral elements (Figure 1, lower panel, right) The upper surface of the ECM box is loaded with an external pressure of 6.0 Pa. The displacement boundary conditions are imposed on the rest of the outer surfaces of the ECM box. The displacements are fixed in perpendicular directions to the surfaces. The ECM can slide in tangent directions to the outer surfaces. The model is continuous; therefore, no contact conditions are imposed.

The cell model consists of nucleus, cytoplasm and cytoskeleton [23]. The current model is extended by employing it in a group of cells embedded in the ECM. The cytoplasm is surrounded by a membrane (cortex). The nucleus and cytoplasm are discretized with tetrahedral finite elements (87,733 and 12,046 elements, respectively). The membranes are modeled using triangular membrane elements (cortex 3236 membranes and nucleus membrane 1792 membranes, Figure 1, upper panel). The cytoskeleton is considered to be a tensegrity structure based on the icosahedron. The deformation of the icosahedral shape is modeled on the observed actual shape of the cell. Prestressed tendons model actin, while the bars model the microtubules. The model consists of six bars and 24 tendons (Figure 1, upper panel).

The model of a group of cells embedded in the ECM consists of 18 cells (Figure 1, lower panel). The number of tetrahedra and triangles in each cell is similar to that of one cell. It slightly varies in each cell due to the meshing algorithm, which is slightly sensitive to the shape of the entire structure.

The geometrical and numerical models are prepared using the GiD v. 17 program. The program is an up-to-date preprocessing tool that prepares the finite element model. The building of the model starts with creating the entire geometry, including the ECM and all the parts of the cells (cortex, cytoplasm, nucleus, membrane surrounding the nucleus and cytoskeleton). The meshing algorithm generates an unstructured mesh with the control of the shape of the elements (their Jacobians, aspect ratio). The mesh is generated as smoothly as possible. If the criteria fail, the mesh is not generated, and the geometry or input parameters must be corrected.

Providing the mesh is generated correctly, which is ensured by the GiD program and the internal controls in the Abaqus program, the influence of the mesh density on the results is not very significant. It has been shown in [24] that the mesh density rather gives smother results, but the maximum values of the field variables are not affected significantly. The denser mesh improves the resolution of the solution. In the case of complex biological living systems, numerical solutions are used for qualitative analysis, which supports the experimentalists.

## 3. Results

### 3.1. The Expression of Stem Cell Markers in Breast Cancer Cell Lines Increases Under 3D Culture Conditions, Regardless of the 3D Model

The experiments were carried out on the two breast cancer cell lines corresponding to luminal cancer (MCF7 and T47D). Standard 2D cultures, 3D floating spheroids on a non-adherent plate and 3D Matrigel cultures were compared for the expression of selected stem cell markers (*SOX2*/SOX2 and *POU5F1*/OCT4) on mRNA and protein levels. 3D cultures (Figure 2A) were collected and processed after 1 and 2 weeks of culture. The qPCR analysis (Figure 2B) demonstrates an increase in both stem cell markers in free floating spheroids vs. 2D culture. Representative immunofluorescence images of both markers in free floating spheroids are presented in Figure 2C, with the quantification of the fluorescent signal in Figure 2D. Western blot analysis of these cultures is shown in Figure 2E. A similar analysis was performed for the 3D matrigel culture of MCF7 cells (Figure 2F,G). The results indicate an increase in both mRNA and protein levels of both stem cell markers in 3D culture versus 2D culture, regardless of the type of 3D culture, although the increase in expression is more evident for SOX2. Interestingly, after 2 weeks of culture, these increased levels of stem cell markers have a tendency to normalize, especially in the case of T47D. The increase in MCF7 is more pronounced and maintains longer than in T47D, thus, this cell line was selected for bioprinting experiments.

### 3.2. The Expression of the Epithelial Marker E-Cadherin Under 3D Versus 2D Conditions Varies Depending on a 3D Model

The expression of the most basic EMT markers, *CDH1*/E-cadherin (epithelial) and *VIM*/vimentin (mesenchymal), was comparatively analyzed in the standard 2D cultures, 3D culture in non-adherent plate and 3D Matrigel cultures. Additionally, E-cadherin expression was assessed in a 3D bioprinted culture, which technically corresponds to the Matrigel type of scaffolded culture. The results indicate that the expression of E-cadherin increases in the culture on a non-adherent plate (Figure 3A–E) but decreases in a scaffold-based culture (both standard Matrigel culture and long-cultured 3D bioprint sample, Figure 3D,E), compared to standard 2D culture. Vimentin expression in the analyzed epithelial breast cancer cell lines is not present, and both analyzed types of 3D culture do not induce it (for reference, the expression of vimentin is shown in the MDA-MB-231 cell line). The second mesenchymal marker was N-cadherin, also not present and not induced in all tested cell lines (Figure 3D).

### 3.3. Bioprinting and Imagining of a 3D Culture for the Generation of the Model

The observed differences in the expression of E-cadherin, the main epithelial marker and the adherent junction protein suggest that the difference could be caused by different tensions between cells in free-floating versus scaffold-based culture. The assumption is that free-floating spheroids should require stronger cell–cell contacts to remain intact, since they are subjected to hydrodynamic forces of the moving fluid and do not have the support of the scaffold. Thus, to evaluate tensions between cells we created a mathematical model of cellular interactions based on the imaging of the 3D bioprinted culture.

The MCF7 cell line was used to generate a mature (8 weeks), self-organized 3D bioprint culture on alginate hydrogel. Out of 34 cultures, five were selected for imaging (Appendix A). Live cells were stained with calcein and Hoechst 33342 to stain live cell cytoplasm and cell nuclei, respectively. The selected image (Culture 1, Figure 4) was used for modeling.

### 3.4. Mathematical Model of Stress Analysis

The numerical model has been prepared with the finite element method, based on previous reports [23]. The parameters assigned for the specific cell’s elements are listed in Table 1. To test the influence of the stiffness of the extracellular matrix on the stress state of cells, two conditions were compared: (1) the assumed Young’s modulus of the ECM, set as 56,200 N/m^2^ (stiff gel) and (2) the assumed Young’s modulus of the ECM, set as 30 N/m^2^ (soft gel). The value for the stiff gel was selected to comply with the stiffness of the alginate hydrogel [25]. The Young’s modulus of the stiff gel is significantly higher than that of the cytoplasm and nucleus, while the Young’s modulus of the soft gel is lower than for the cytoplasm and the nucleus.

#### 3.4.1. ECM Stress

Cell in ECM is interpreted as an inclusion in continuous medium. The maximum HMH stress is close to the top and bottom of the cell (inclusion). In the case of the stiff gel, the region of high stress is concentrated around the entire cell; similar concentration is observed for soft gel, but the stress gradients are milder. To enhance the differences present in the linear scale (Figure 5A,C), a decimal logarithm scale is used (Figure 5B,D). Figure 5 presents HMH stress distribution in ECM around one cell calculated for the stiff (Figure 5A,B) and soft gel (Figure 5C,D). The stress is higher in the case of the stiff ECM than in the case of the soft ECM.

HMH stress distribution in the vertical cross-section of the matrix for a group of cells is shown in Figure 6. The maximum stress is higher in the stiff matrix than in the soft one. When comparing single-cell and group-cell cases, the maximum stress in ECM is lower in the single-cell. This statement is valid for both types of ECM materials, stiff and soft ECM. The maximum HMH stress is given in Table 2.

#### 3.4.2. Cortices

The HMH stress distributions in the cortex of a single cell case and the cortices of the group of cells have been calculated. The HMH stress field in the single cell in the case of stiff ECM is qualitatively different from the field in soft ECM (Figure 7). HMH stress is distributed relatively smoothly in the case of the single cell immersed in the stiff gel. The highest stress is 2.16 times higher than the lowest, while in the case of the soft ECM, the highest stress is 8.25 times higher than the lowest. The stress in the case of the soft ECM characterizes high punctual stress concentrations. They are located around the points where the cytoskeletal nodes are attached to the membranes.

Maximum stress in the cortex is higher for soft gel, in both the single-cell case and the group-of-cell case values (Table 3). However, the qualitative picture of the stress distribution is similar. The stress concentrations are around the points of attachment to the cytoskeleton. The stress distribution is relatively smooth within the rest of the surface (Figure 8A,B).

Figure 8C illustrates the interaction between cells via ECM. The figure is prepared by uncovering the inner cells. For clarity, the cells do not touch each other, with a small gap between them. The decimal logarithmic scale is used to improve stress concentrations. Stress concentrations due to attachments of the cytoskeleton in the form of small dots are visible in both cases of ECM. However, in the case of stiff ECM, high-stress islands are visible close to the equator of the cells. In the case of soft ECM, the stress gradients are lower, and the stress field is much smoother.

#### 3.4.3. Cytoplasm

As in the cortices, the maximum stress in the cytoplasm is higher in the case of soft EMC than in the case of stiff ECM (Table 4). A qualitative comparison of the stress fields in the single cell revealed that in the case of both ECMs, the stress concentration appeared in the cytoskeleton attachment sites (Figure 9A). The vertical cross-sections of the cell show that in the case of stiff ECM, the stress concentration appears around the nucleus, which plays the role of inclusion into the cytoplasm. In the case of soft ECM, stress is significantly higher than in the case of stiff ECM, but the stress concentration around the nucleus is milder than in the case of stiff ECM (Figure 9B, enhanced in Figure 9C).

Figure 10A,B shows the stress distribution in the cytoplasm of a group of cells. Stress fields in nuclei demonstrate stress concentrations in places of cytoskeleton attachments. This is similar to the single-cell case. However, the stress distributions are qualitatively different considering the cases of stiff and soft ECM. A strong interaction via ECM can be noted in the case of stiff ECM. High stress is seen in the outer surfaces of the cytoplasm (Figure 10B) and close to the surfaces of the cytoplasm (Figure 10C). The stress distribution is different in the soft and stiff ECM (Figure 10B). Horizontal cross sections show high stress close to the centers of the cytoplasm (Figure 10C), opposite to the stiff ECM case. The cells interact with each other since the stress fields in the cytoplasm are not homogeneous.

#### 3.4.4. Nuclei Membranes

HMH stress in nucleus membrane in the single cell case is depicted in Figure 11. The maximum HMH stress is significantly higher in the case of soft ECM than in the case of stiff ECM (Table 5). The HMH stress distribution in the membranes is qualitatively different in the case of stiff ECM from the case of the soft ECM. In the case of stiff ECM, the HMH stress is lower, close to the equator of the cell than close to the poles. In the case of soft ECM, the stress is low, close to the poles.

For the group of cells, the HMH stress in the nuclei membranes (Figure 12) is significantly lower in the case of stiff ECM (upper panel) than in the case of soft ECM (lower panel). The distribution of stress in the case of stiff ECM is similar in both layers of the cells. In contrast, in the case of soft ECM, HMH stress is higher in the lower layer of the cells than in the upper one (Figure 12).

#### 3.4.5. Nuclei

The stress fields in the nucleus in the single-cell case are shown in Figure 11. Maximum HMH stress is much more prominent in the case of soft ECM than in the case of stiff ECM (Table 6). The stress distribution is qualitatively different when comparing the two cases (Figure 11 upper panel vs. lower panel). In the stiff ECM, higher stress is present at the bottom of the nucleus, whereas, in the soft ECM, higher stress is present at the upper part of the nucleus. The stress distribution in the vertical cross-sections confirms the latter.

HMM stress distribution in the group of nuclei in the cells is depicted in Figure 12. Maximum stress is lower in the case of stiff ECM (upper panel) than in the case of soft ECM (lower panel). The qualitative evaluation of the stress distribution leads to the same conclusion as the estimation in the nuclei membranes.

#### 3.4.6. Cytoskeletons

The displacements and stress distribution on the cytoskeleton in the single cell are shown in Figure 13. The displacements are significantly higher in the case of the soft ECM than in the case of the stiff ECM (Figure 13A). This is mostly the motion against a z-direction. The uniaxial stress in a cytoskeleton in the single cell is shown in Figure 13B. The absolute value of stress is higher in the case of the soft ECM than in the case of the stiff ECM (minus sign means compressive stress, plus sign means tensile stress).

Figure 14 shows the behavior of the cytoskeletons for the group of cells. The displacements in the case of stiff ECM are significantly lower than in the case of the soft ECM (Figure 14A). The displacements of the upper layer of the cells are smaller than those of the lower layer. However, in the case of soft ECM the relative difference in the displacements is more distinct. The upper layer moves almost in a homogeneous manner. The movement of the lower layer is much less distinct than that of the upper layer.

The absolute value of the tensile and compressive stress is significantly higher in the case of soft ECM than in the case of stiff ECM (Figure 14B, Table 7).

## 4. Discussion

Different growth conditions affect cell genotypes and phenotypes. Variations in gene and protein expression between 2D and 3D cultures have been repeatedly reported, along with differences in the response to drugs [30,31,32,33].

The increase in stem cell marker expression has already been observed under 3D conditions in MCF7 cells grown on collagen scaffolds [3,34]. The authors also reported the upregulation of mesenchymal markers and the downregulation of E-cadherin (*CDH1*) in these conditions, which was confirmed by other reports [6,35]. Interestingly, for cells grown in a different 3D model (ultra-low attachment plate), an increase in both *SOX2* and *CDH1* expression was reported [7].

The plasticity of the expression of E-cadherin observed in our study conforms to these previous reports, although the authors of these reports always analyzed only one 3D model and did not provide a comparison of different 3D models as we do in this report. Qi et al. [36] compared scaffold-free and scaffold-based spheroids in their report, but they analyzed NSCLC cells and compared different markers. While we do not exclude the possibility that the epithelial–mesenchymal transition is engaged in a scaffold-based culture, it is obviously not the case for a scaffold-free culture on a low-adherent plate. Accordingly, since we observe dramatic differences in E-cadherin levels between the two studied 3D models (scaffold-free versus scaffold-based), 3D conditions per se cannot explain these differences, and we propose the solution based on mechanobiology.

The main difference between the analyzed 3D models consists in the physical conditions of growth, mainly in the fact that on the non-adherent plate cells grow in the medium, with a relative freedom of movement, while on the scaffold they rest in the stiff hydrogel and have restricted movement. We assumed that free-floating spheroids require stronger cell–cell contacts (hence: E-cadherin expression) to remain intact, since they are subjected to hydrodynamic forces of the moving fluid and do not have the support of the scaffold. Some reports support this reasoning: for example, Buckley et al. [37] suggested that mechanical tension reinforces the stability of cadherin-based cell-cell junctions, while Verma et al. [38] observed the same effect during the flow. To test this hypothesis, we simulated mechanical stress within the single cell and between the group of cells, using a mathematical model, based on the confocal images of a bioprinted culture.

The model has been used to compare two different conditions: with a very stiff matrix mimicking an alginate hydrogel and a very soft matrix to approximate (although not completely) the conditions in the medium. The results for every cellular element (cortex, cytoplasm, nucleus, cytoskeleton) and for a single cell, as well as for a group of cells, demonstrate that the stress, especially the maximal stress, is always higher in the conditions of soft gel. Although there are interesting differences between the stress field distribution for stiff and soft gel, the highest values for soft gel, especially at the attachment sites, support our hypothesis that higher stress requires stronger cell–cell junctions to preserve the integrity of the group. If not for the upregulation of E-cadherin (or possibly other junction proteins), this group of cells would disintegrate into single cells, which in 3D conditions would be detrimental to their survival. This may be the reason for the reported upregulation of some junction proteins in circulating tumor cell clusters [39,40] and have profound significance for the metastatic process.

The other important aspect of the observed differences between the two analyzed 3D models is that in cancer research, along with the growing awareness of the unreliability of 2D models, many drug-testing platforms use 3D models, mostly selected for their reproducibility and general feasibility. Researchers performing these experiments should be aware of the differences between the 3D models used, while interpreting the results. Furthermore, several studies have already demonstrated that ECM stiffness affects cancer cell resistance [41,42,43]. This study may contribute to the explanation of these differences.

## 5. Conclusions

This work confirms and expands the knowledge about the apparent differences between the expression of stem cell and EMT markers under 2D and 3D growth conditions, but it also highlights the differences between 3D models that were not previously reported, namely E-cadherin plasticity in scaffold-free and scaffold-based conditions. To explain this plasticity on the ground of mechanobiology, we have generated a mathematical model of a group of cells and tested the stress on cellular elements in relation to the adjusted stiffness of the ECM. The results confirm the hypothesis that the higher stress is observed in a soft matrix, approximating scaffold-free conditions, which requires stronger cell–cell junctions to preserve the integrity and the survival of a group of cells. These conclusions are significant for the theoretical importance of the behavior of cell clusters in the metastatic process as well as for practical purposes, for example, during the analysis of different drug-testing panels.

## Figures and Tables

**Figure 1 bioengineering-12-00147-f001:**
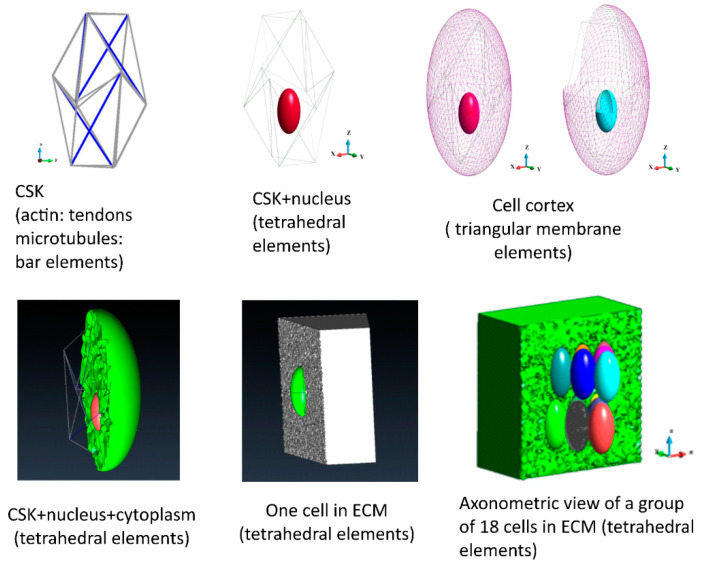
The geometry of the elements of the model; cytoskeleton (CSK), nucleus, cell cortex, cytoplasm, ECM.

**Figure 2 bioengineering-12-00147-f002:**
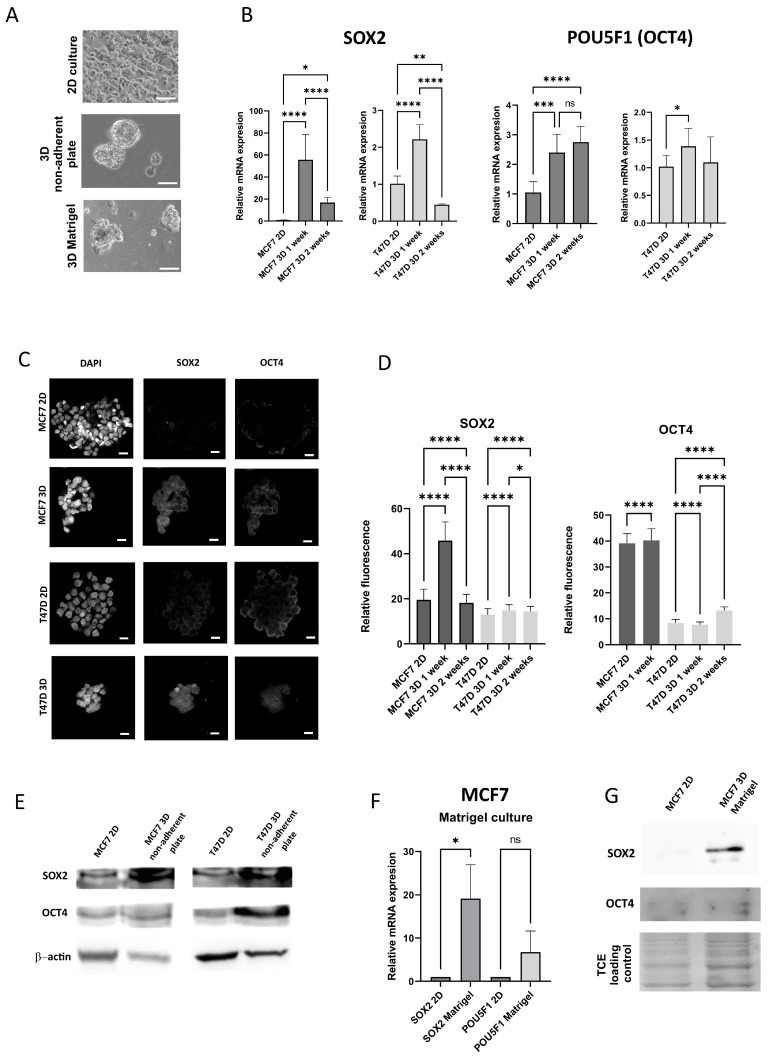
Expression of stem cell markers under 2D and 3D conditions. (**A**) Representative images of 2D and 3D cultures, MCF7 cells, Olympus CKX53, scale bar 20 μm. (**B**) qPCR results of SOX2 and POU5F1 expression in 2D and 3D (non-adherent plate) conditions, MCF7 and T47D cell lines, (**C**) Representative confocal images of the 2D culture and 1-week 3D culture on non-adherent plate. Scale bar: 20 μm (**D**) Quantification of the fluorescence for SOX2 and OCT4, 2D and 3D on non-adherent plate. (**E**) Western blots for SOX2 and OCT4, 2D and non-adherent plate, (**F**) qPCR results for the culture on Matrigel, (**G**) Western blot results for the culture on Matrigel. *p*-values: *—<0.05, **—<0.01, ***—<0.001, ****—<0.0001, ns: non-significant.

**Figure 3 bioengineering-12-00147-f003:**
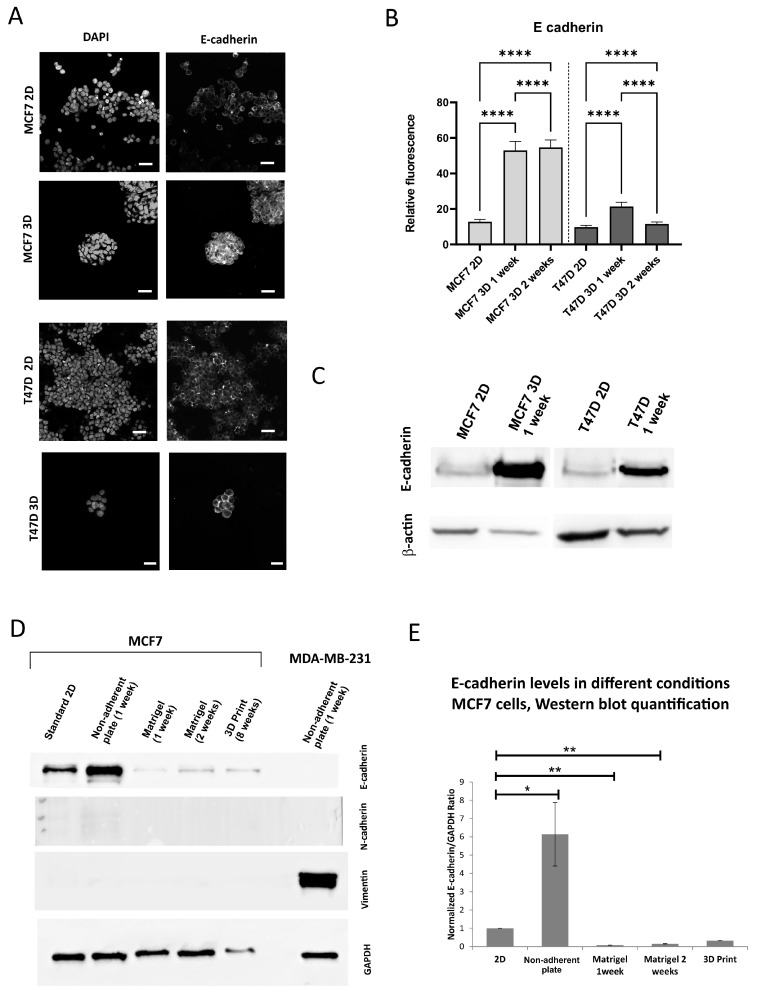
Expression of E-cadherin under 2D and 3D conditions. (**A**) Representative confocal images of the 2D culture and 1 week 3D culture on non-adherent plate, scale bar 20 μm. (**B**) Quantification of E-cadherin fluorescence, 2D and 3D on non-adherent plate, *p*-values: <0.0001. (**C**) Western blots for E-cadherin, 2D and 3D on a non-adherent plate, (**D**) Representative Western blot for epithelial and mesenchymal markers, 2D, 3D on a non-adherent plate, 3D in Matrigel, bioprinted 3D culture in alginate, (**E**) Quantification of Western blot experiments, 3–5 repeats, *p*-values, respectively: 0.0239, 0.0036, 0.0074. *p*-values: *—<0.05, **—<0.01, ****—<0.0001.

**Figure 4 bioengineering-12-00147-f004:**
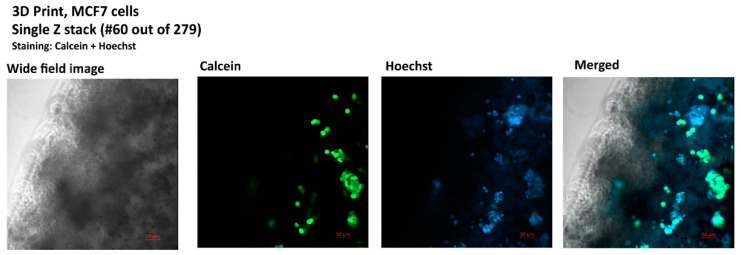
Image of a single Z-stack from a 279, confocal image, bioprinted MCF7 culture, staining: calcein (live cells) Hoechst 33342 (nuclei), Zeiss LSM800 microscope (Carl Zeiss AG, Oberkochen, Germany).

**Figure 5 bioengineering-12-00147-f005:**
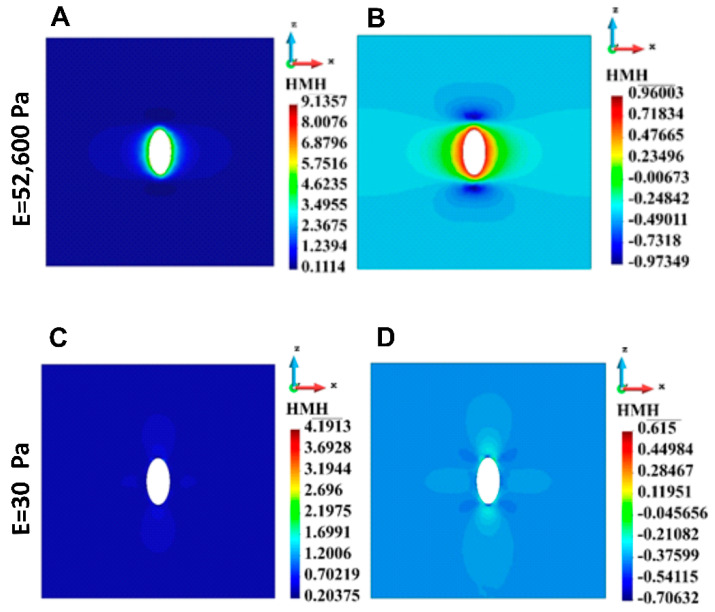
HMH stress distribution in the cross-section across the ECM with single cell. (**A**) Linear scale, stiff ECM, (**B**) Logarithmic scale, stiff ECM, (**C**) Linear scale, soft ECM, (**D**) Logarithmic scale, soft ECM.

**Figure 6 bioengineering-12-00147-f006:**
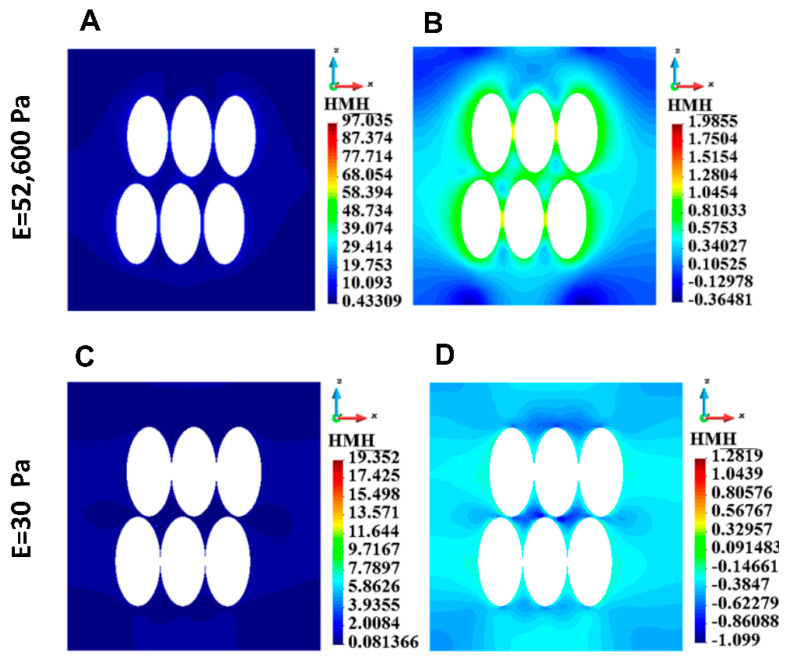
HMH stress distribution in the ECM for a group of cells, stiff ECM (upper panel) and soft ECM (lower panel). (**A**) Linear scale, stiff ECM, (**B**) Logarithmic scale, stiff ECM, (**C**) Linear scale, soft ECM, (**D**) Logarithmic scale, soft ECM.

**Figure 7 bioengineering-12-00147-f007:**
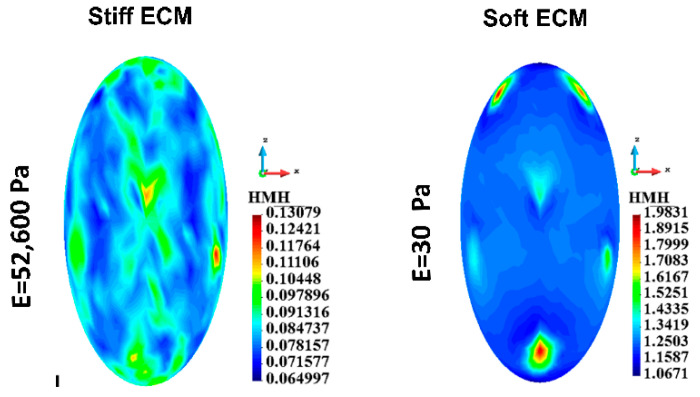
Single cell, cortex, HMH stress; Stiff ECM, Soft ECM.

**Figure 8 bioengineering-12-00147-f008:**
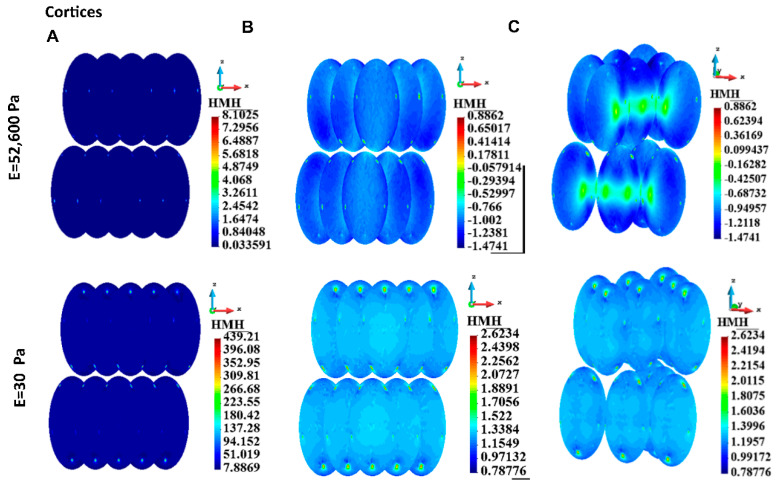
HMH stress in cortices for a group of cells, stiff ECM (top panel), and soft ECM (lower panel). (**A**) Linear scale, (**B**) Logarithmic scale, (**C**) Illustration of interactions between cells.

**Figure 9 bioengineering-12-00147-f009:**
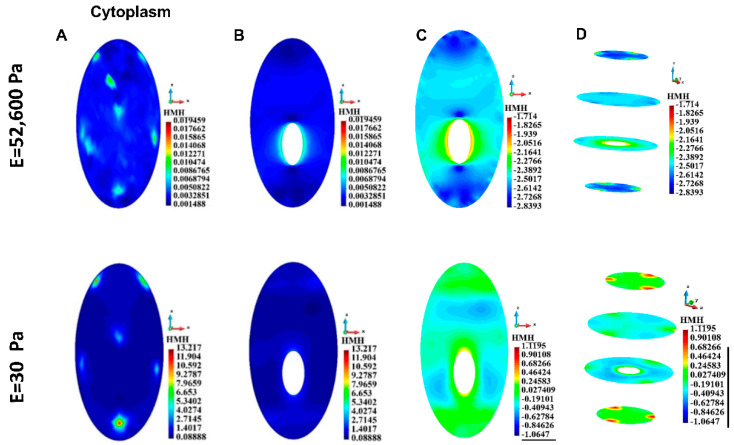
HMH stress distribution, single cell, cytoplasm. (**A**) Cell surface. (**B**) Vertical cross-section of the cytoplasm, linear scale. (**C**) Vertical cross-section of the cytoplasm, decimal logarithm scale. (**D**) Horizontal section of the cytoplasm, decimal logarithm scale.

**Figure 10 bioengineering-12-00147-f010:**
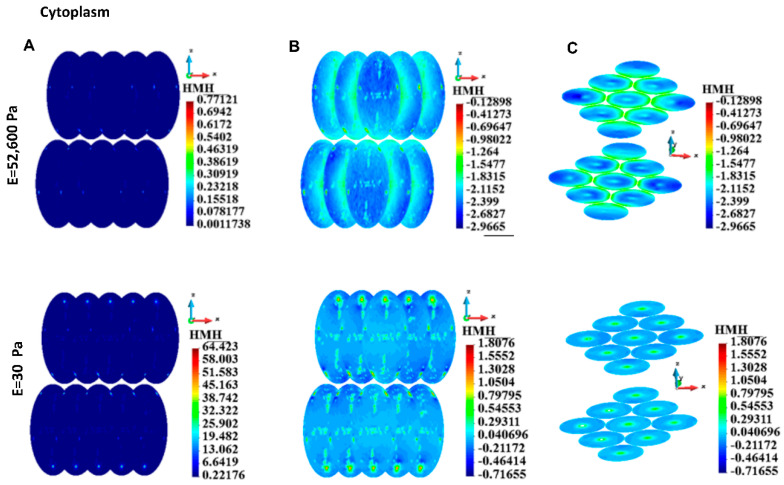
HMH stress, group of cells, cytoplasm. (**A**) Linear scale, (**B**) Side view, decimal logarithm scale, (**C**) Horizontal cross-section, axonometric view, decimal logarithm scale.

**Figure 11 bioengineering-12-00147-f011:**
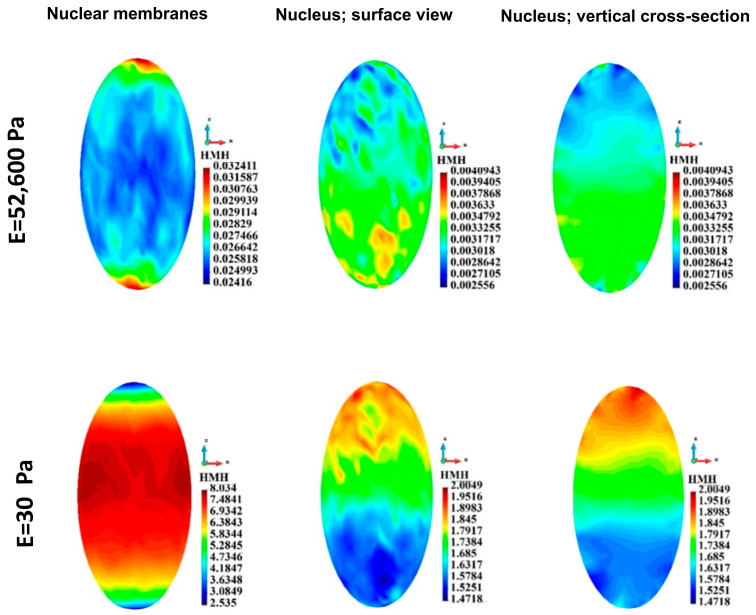
HMH stress distribution in a single nuclear membrane and the nucleus. Stiff (upper panel) and soft (lower panel) ECM.

**Figure 12 bioengineering-12-00147-f012:**
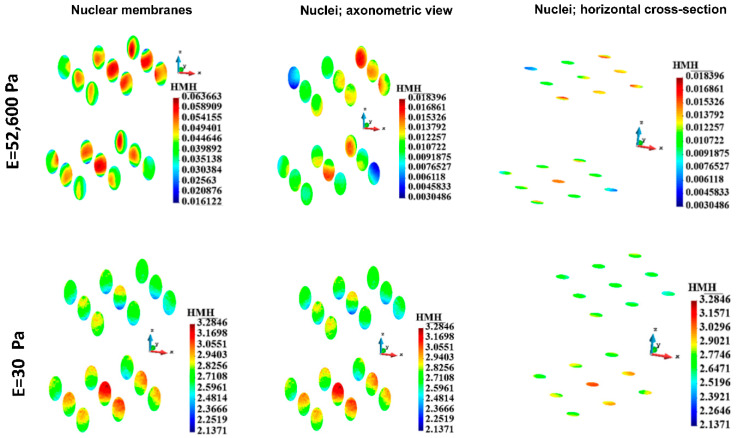
HMH stress distribution in nuclear membranes and nuclei for the group. Stiff (upper panel) and soft (lower panel) ECM.

**Figure 13 bioengineering-12-00147-f013:**
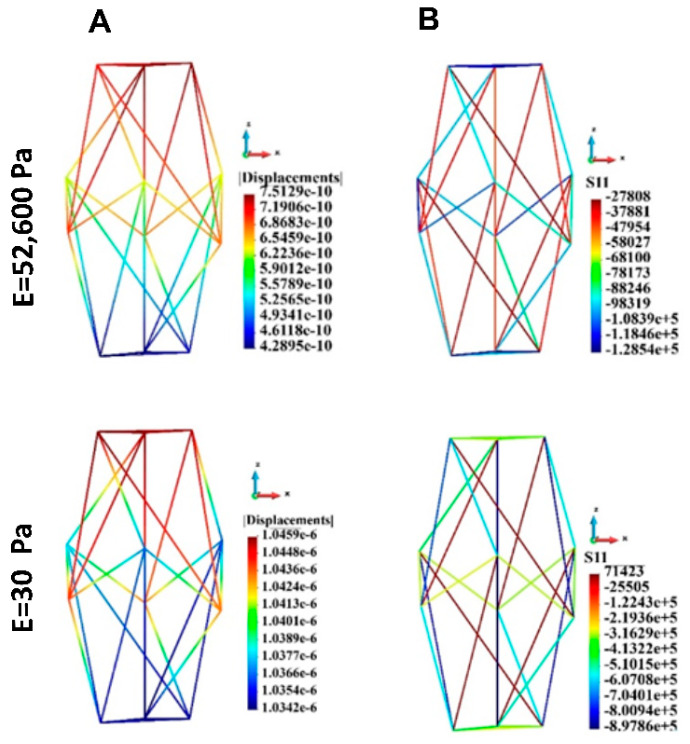
Cytoskeleton, single cell; displacements and uniaxial stress; (**A**) Displacements, (**B**) uniaxial stress. Stiff (upper panel) and soft (lower panel) ECM.

**Figure 14 bioengineering-12-00147-f014:**
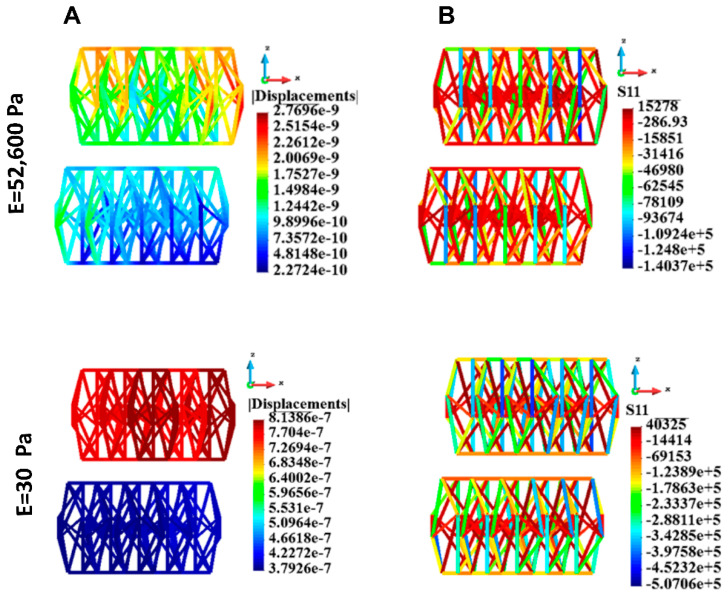
Cytoskeleton, group of cells; displacements and uniaxial stress; upper panel: stiff ECM, lower panel: soft ECM. (**A**) Displacements, (**B**) Uniaxial stress. Stiff (upper panel) and soft (lower panel) ECM.

**Table 1 bioengineering-12-00147-t001:** Model parameters assigned to specific elements.

Cell Element	Young Modulus (E) (Pa)	Poisson Ratio (v)	Area (m^2^)	Thickness (m)
Actin [26]	2.6 × 10^9^		18.0 × 10^−18^ [23]	
Microtubules [26]	1.2 × 10^9^		190.0 × 10^−18^ [23]	
Cytoplasm [27]	100.0	0.37 [28]		
Nucleus [29]	400.0	0.37 [28]		
Cortex	1000.0	0.3		6.0 × 10^−9^
Nucleus membrane	1000.0	0.3		6.0 × 10^−9^
ECM	56,200 (stiff) 30.0 Pa (soft)	0.4777		

**Table 2 bioengineering-12-00147-t002:** Maximum HMH stress in ECM.

Case	Stiff ECM (Pa)	Soft ECM (Pa)
1 cell	9.14	4.19
18 cells	97.03	19.35

**Table 3 bioengineering-12-00147-t003:** Maximum HMH stress in cortices.

Case	Stiff ECM (Pa)	Soft ECM (Pa)
1 cell	0.13	96.60
18 cells	8.13	439.21

**Table 4 bioengineering-12-00147-t004:** Maximum HMH stress in the cytoplasm.

Case	Stiff ECM (Pa)	Soft ECM (Pa)
1 cell	0.02	13.22
18 cells	0.77	64.24

**Table 5 bioengineering-12-00147-t005:** Maximum HMH stress in nuclei membranes.

Case	Stiff ECM (Pa)	Soft ECM (Pa)
1 cell	0.032	8.034
18 cells	0.063	3.28

**Table 6 bioengineering-12-00147-t006:** Maximum HMH stress in the nuclei.

Case	Stiff ECM (Pa)	Soft ECM (Pa)
1 cell	0.04	2.00
18 cells	0.018	3.28

**Table 7 bioengineering-12-00147-t007:** Maximal and minimal uniaxial stress.

Case	Stiff ECM		Soft ECM	
	S11 Max	S11 Min	S11 Max	S11 Min
1 cell	27,808.0	−1.28 × 10^5^	71,423.0	−8.98 × 10^5^
18 cells	15,278.0	−1.40 × 10^5^	40,325.0	−5.07 × 10^5^

## Data Availability

Data are contained within the article and Appendix A.

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
