# Peer review of "Plasticity of Expression of Stem Cell and EMT Markers in Breast Cancer Cells in 2D and 3D Culture Depend on the Spatial Parameters of Cell Growth; Mathematical Modeling of Mechanical Stress in Cell Culture in Relation to ECM Stiffness"

_bioengineering, 2025, doi:10.3390/bioengineering12020147_

Round 1
Reviewer 1 Report
Comments and Suggestions for Authors
This study is interesting, related to mathematical modeling of stress in cell culture. There is a well-structured organization in the manuscript while presenting interesting results. I suggest publication after revisions according to the below comments.
· Language should be checked.
· Introduction must be enhanced by discussing the current technology in the field.
· Novelty must be given clearly.
· Authors must discuss what kind of developments could be achieved by the outputs obtained in this study.
· How did authors obtain the materials data for numerical model?
· Is there a preliminary study for model optimization, such as mesh size optimization, etc.?
· Modeling must be given in detail. Time scheme procedure, boundary conditions, contact types, etc.?
· Authors must give details about validation of FEA results.
· There must be a section “Conclusions”.
Author Response
Response to Reviewer 1
This study is interesting, related to mathematical modeling of stress in cell culture. There is a well-structured organization in the manuscript while presenting interesting results. I suggest publication after revisions according to the below comments.
1· Language should be checked.
The language was corrected.
2· Introduction must be enhanced by discussing the current technology in the field.
The Introduction was corrected, with more insight into the current techniques and their limitations.
3· Novelty must be given clearly.
In the Introduction and Discussion/Conclusions of the revised manuscript the novelty of the results was clearly presented.
4· Authors must discuss what kind of developments could be achieved by the outputs obtained in this study.
In the Discussion/Conclusions of the corrected manuscript the potential applicability of these results in cancer research was discussed.
5· How did authors obtain the materials data for numerical model?
The material data was collected based on available literature. Table 1 is updated with the appropriate references.
6· Is there a preliminary study for model optimization, such as mesh size optimization, etc.?
The meshing process is strongly automated. It is due to the use of the up-to-date preprocessing tool which is the preprocessor GiD. The building of the model starts with the creation of the entire geometry including the ECM, and all the parts of the cells (cortex, cytoplasm, nucleus, membrane surrounding the nucleus and the cytoskeleton). The meshing algorithm generates unstructured mesh with the control of shape of the elements (their Jacobians, aspect ratio, etc.).
It has been found the size of the mesh providing the mesh is correct does not affect significantly the solution. The mesh density is rather a matter of the resolution of the solution. It is shown in the paper:
Postek E., Sadowski T., Thermomechanical effects during impact testing of WC/Co composite material, COMPOSITE STRUCTURES, DOI: 10.1016/j.compstruct.2020.112054 , Vol.241, pp.112054-1-25, 2020.
In particular, the effects are shown in Figure 15 (mesh definition) and in Table 6 (Mesh dependency check).
The finite element system is solved in about 1 hour using 48 cpus.
7· Modeling must be given in detail. Time scheme procedure, boundary conditions, contact types, etc.?
The statements about the time scheme procedure and boundary conditions are enhanced in the text. It is a continuous model. There are no contact conditions. We mentioned and marked in the text.
8· Authors must give details about validation of FEA results.
We refer to the published results of the model which we extended an modified.
McGarry, J.G.; Prendergast, P.J. A three-dimensional finite element model of an adherent eukaryotic cell. Eur Cell Mater 2004, 7, 27-33; discussion 33-24, doi:10.22203/ecm.v007a03.
In the model that is presented in the reference above, a single cell was investigated. In our case the model is extended towards modelling of a group of cells embedded in extracellular matrix. The single cell model considered in the reference was analyzed using Ansys. To the extent that the authors of the reference and we used, Abaqus program gives the same results.
The authors of the reference used similar mesh density (it is to check visually, they did not give the exact values) to our single cell meshes. In consequence, we can be sure of the correctness of the results providing the meshes are correct. It is within the assumptions and limitations of the model.
We would like to enhance that the numerical solutions are used in qualitative analysis of the system of cells. It is due to significant simplifications of such a complex system like the living matter.
In addition, we can point out a model of a cytoskeletal network as follows:
Postek, E. Parameter sensitivity of a monolayer tensegrity model of tissues. J Coupled Syst Multiscale Dyn 2014, 2, 179-187, doi:10.1166/jcsmd.2014.1054.
9 There must be a section “Conclusions”.
A section “Conclusions” has been included.

Reviewer 2 Report
Comments and Suggestions for Authors
1. The abstract does not clearly demonstrate how the findings contribute new insights to the field of cancer research, as many studies have already discussed the limitations of 2D and animal models.
2. The main objective of the study is not explicitly stated in the abstract, making it challenging to understand the focus of the research.
3. Coherence and flow are key concepts that need improvement in the abstract.
4. The introduction lacks proper organization and is notably brief, containing only a limited number of references. Additionally, the identification and explanation of the research gap are insufficiently developed. It would greatly enhance the clarity of the paper if the main objective were articulated more clearly and concisely at the end of the introduction section. Including a more comprehensive review of relevant literature and a more explicit statement of the research gap would provide a stronger foundation for the study.
5. All abbreviations should be defined upon their first use to enhance clarity and make the content more accessible to a wider audience. (such as: MCF7 (ATCC), T47D (DSMZ) and MDA-MB-231 (ATCC), along with any others used throughout the text.).
6. The listing of secondary antibodies includes repeated entries (e.g., Anti-Mouse IgG-AlexaFluor488), which suggests a lack of attention to detail and can confuse readers.
7. The imaging section lacks detail about the settings or parameters used on the Zeiss Axio observer Z1 LSM 800 confocal microscope, which are essential for reproducibility.
8. The choice of harvesting cells on the 7th day of 3D culture and at ~80% confluency in 2D culture is not explained, which raises questions about the rationale behind these specific time points.
9. The section on cDNA synthesis lacks detail regarding the conditions used (e.g., temperature, time), which are important for understanding the process.
10. The choice of BIOINK hydrogel is mentioned but not justified. An explanation of why this specific hydrogel was selected would be beneficial.
11. The mention of the Abaqus program and GiD program requires more detail about their specific roles in the modeling process and why they were chosen.
12. The mathematical formulations are missing important references, definition of variables and should be formatted correctly using MathType or an alternative method.
13. Section 8.8.2: The authors mentioned that the meshing algorithm is sensitive to the shape of the structure but does not explain how this affects the modeling process or results. Add further explanations to figure 1.
14. Has any mesh quality analysis been carried out on the proposed FE model (aspect ratio, Jacobian rate, etc.)? Please justify.
15. While it is stated that MCF7 was selected for bioprinting experiments, there is no discussion on why this choice was made in relation to T47D.
16. While the results regarding E-cadherin and vimentin expression are stated, there is no detailed interpretation of what these changes imply for the biology of the cell lines studied.
17. Improve the resolution of Figures 2, 3 and 4.
18. The boundary conditions of the finite element model should be specified more clearly within the model itself.
19. The statement that "the maximum stress in cytoplasm is higher in the case of soft ECM" lacks context. Explaining why this difference matters in biological terms would enhance understanding.
20. It is essential to include a conclusion at the end of the manuscript.
21. The manuscripts are excessively lengthy, containing numerous redundancies and failing to maintain a clear focus throughout. This can make it challenging for readers to grasp the main points and key findings effectively. Streamlining the content and eliminating repetitive information would enhance clarity and engagement.
Author Response
Response to Reviewer 2
- The abstract does not clearly demonstrate how the findings contribute new insights to the field of cancer research, as many studies have already discussed the limitations of 2D and animal models.
The abstract was corrected, highlighting the importance of the study for the proper interpretation of the results obtained using a specific 3D model for cancer research.
- The main objective of the study is not explicitly stated in the abstract, making it challenging to understand the focus of the research.
The abstract was corrected and clarified.
- Coherence and flow are key concepts that need improvement in the abstract.
The abstract was corrected and clarified.
- The introduction lacks proper organization and is notably brief, containing only a limited number of references. Additionally, the identification and explanation of the research gap are insufficiently developed. It would greatly enhance the clarity of the paper if the main objective were articulated more clearly and concisely at the end of the introduction section. Including a more comprehensive review of relevant literature and a more explicit statement of the research gap would provide a stronger foundation for the study.
The introduction was corrected, expanded and clarified, with a more comprehensive review of the literature. The explanation of the research gap was provided.
- All abbreviations should be defined upon their first use to enhance clarity and make the content more accessible to a wider audience. (such as: MCF7 (ATCC), T47D (DSMZ) and MDA-MB-231 (ATCC), along with any others used throughout the text.).
The abbreviations were defined were possible, although some are just given names (e.g. cell lines names), not abbreviations.
- The listing of secondary antibodies includes repeated entries (e.g., Anti-Mouse IgG-AlexaFluor488), which suggests a lack of attention to detail and can confuse readers.
The list was corrected.
- The imaging section lacks detail about the settings or parameters used on the Zeiss Axio observer Z1 LSM 800 confocal microscope, which are essential for reproducibility.
The details were provided.
- The choice of harvesting cells on the 7th day of 3D culture and at ~80% confluency in 2D culture is not explained, which raises questions about the rationale behind these specific time points.
The time points for cell harvest were associated with the spheroid formation process. Early-phase spheroids are defined as spheroids in 5-7 day of culture, and most probably it is also the peak of the expression for stem cell markers - which was confirmed in our preliminary research. Other time points (e.g. 14 days for mature spheroids) were planned according to biological nature of the model, considering the maturation and formation of the cell aggregates. The harvest of the 2D model was set according to standard procedures considering GLP cell culture.
9. The section on cDNA synthesis lacks detail regarding the conditions used (e.g., temperature, time), which are important for understanding the process.
The conditions for the PCR reaction were provided.
- The choice of BIOINK hydrogel is mentioned but not justified. An explanation of why this specific hydrogel was selected would be beneficial.
We selected the BIOINK hydrogel based on alginate due to its well-documented physicochemical properties and widespread application in 3D bioprinting (10.1021/acsabm.3c00984, 10.3390/md21030177, 10.1039/c7bm00765e). Alginate-based hydrogels are known for their mechanical and structural stability, ensuring reproducible experimental results—particularly important in studies involving mathematical modeling. Additionally, the selected hydrogel demonstrates high biocompatibility, creating a cell-friendly microenvironment for long-term cell cultures in 3D structures, as previously documented by our team (Kurzyk A et al 2024 Biofabrication 16 025021) making it a suitable The justification for selecting the BIOINK hydrogel has been added to the revised manuscript in Section 2.7.
- The mention of the Abaqus program and GiD program requires more detail about their specific roles in the modeling process and why they were chosen.
It was explained and marked in the text.
- The mathematical formulations are missing important references, definition of variables and should be formatted correctly using MathType or an alternative method.
The variables were all defined in the text. The formatting of the equations was done with the current version of MathType. Unfortunately, the most recent versions of MathType do not give very good results. Now, it is done using the embedded in the Word 365 equation editor with latex compatible fonts. We trust it improves the outcome. Of course, Latex is the best, but the requirement is to use Word. We hope it is sufficient for the five equations. We believe that the technical editor will evaluate it and give a good advice.
We added one reference concerning the strain increment (linear and nonlinear part), a textbook by Y.C. Fung, Foundations of Solid Mechanics, 2nd Edition, Prentice Hall, 1965.
- Section 8.8.2: The authors mentioned that the meshing algorithm is sensitive to the shape of the structure but does not explain how this affects the modeling process or results. Add further explanations to figure 1.
We mentioned “it is slightly sensitive”. It does not affect the solution significantly. Providing the mesh is correct, the results are not significantly affected by the mesh density. The mesh density is rather a matter of the resolution of the solution. It is shown in the paper:
Postek E., Sadowski T., Thermomechanical effects during impact testing of WC/Co composite material, COMPOSITE STRUCTURES, DOI: 10.1016/j.compstruct.2020.112054 , Vol.241, pp.112054-1-25, 2020.
In particular, the effects are shown in Figure 15 (mesh definition) and in Table 6 (Mesh dependency check).
It worth to note the numerical solutions are used in qualitative analysis of the system of cells. It is due to inherent simplifications of such a complex system like the living matter.
More explanations concerning meshing are given below (point 14).
- Has any mesh quality analysis been carried out on the proposed FE model (aspect ratio, Jacobian rate, etc.)? Please justify.
The process of mesh creation starts from building the entire geometry of the system. The geometry is built with GiD preprocessing program (https://www.gidsimulation.com/ ). The program is well tested und used in several industries. The unstructured meshing tool checks the correctness of the elements concerning the basic features like Jacobian and aspect ratio. In the case, the values are incorrect, the mesh is not generated.
The next controlling steps are associated with the program Abaqus. The control of the solution is done at the preprocessing stage and during the finite element system of equations solution phase. During the preprocessing phase, the elements are checked with respect to the mentioned features. In the case of inacceptable parameters, the solution is terminated. It means, the mesh has to be corrected. The failing elements are given in the output from the preprocessing phase.
The resulting FE equations system is checked during the solution phase as well. In particular, it is tested for low Jacobian values. If the solution is not well posed it can be terminated or the number of iterations is increased (an artificial stiffness is added to the system, there are warnings due to the fact).
These issues occurred during the mesh calibration phase. During the final phase, the above mentioned issues did not occur.
- While it is stated that MCF7 was selected for bioprinting experiments, there is no discussion on why this choice was made in relation to T47D.
The MCF-7 cells line was chosen for 3D printing model because of its excellent spheroid-forming properties. The 3D printing model was done after the spheroid culture analysis. The MCF-7 cell line formed most replicable spheroids and was found to have highest and longer maintaining SOX2 peak during the spheroid culture, which was the reason for selecting this cell line for further experiments (as stated in the manuscript in section 3.1.).
- While the results regarding E-cadherin and vimentin expression are stated, there is no detailed interpretation of what these changes imply for the biology of the cell lines studied.
MCF7 and T47D are the standard, commercially available, well characterized, epithelial cell lines with high E-cadherin expression and no vimentin expression, since it is a mesenchymal marker. The rationale for these studies was that one of the goals of our experiments was to observe if there are any signs of EMT, or partial EMT occurring in 3D culture. The EMT and the expression of its markers was described many times and we did not want to repeat this argumentation, although, for clarity, we added some references We did not observe significant changes in vimentin levels, so this is not discussed. However, we feel that changes in E-cadherin expression were discussed extensively in the manuscript, because it is the main point of the results.
- Improve the resolution of Figures 2, 3 and 4.
The tiff documents used to generate a pdf file have a resolution of 600 dpi, which is a standard quality. We can improve the quality, if needed, but maybe providing the original tiff files will be enough.
- The boundary conditions of the finite element model should be specified more clearly within the model itself.
It has been explained and marked in the text.
- The statement that "the maximum stress in cytoplasm is higher in the case of soft ECM" lacks context. Explaining why this difference matters in biological terms would enhance understanding.
The maximal stress was calculated for all cellular elements. While the stress on cortices and cytoskeletal elements is more relevant to the explanation of the differences in E-cadherin expression, the stress on other cellular elements also should be reported, as a part of the comprehensive analysis. In this case, higher stress associated with soft ECM complies with all other results. Additionally, a part of the Discussion section in which the effect of mechanical stress on cell junctions and cytoskeleton is commented has been expanded in a revised version of the manuscript.
- It is essential to include a conclusion at the end of the manuscript.
A section ‘Conclusions’ has been included.
- The manuscripts are excessively lengthy, containing numerous redundancies and failing to maintain a clear focus throughout. This can make it challenging for readers to grasp the main points and key findings effectively. Streamlining the content and eliminating repetitive information would enhance clarity and engagement.
In the corrected version we tried to eliminate redundancies and improve clarity.
Reviewer 3 Report
Comments and Suggestions for Authors
In this manuscript, the authors have analyzed the plasticity of expression of stem cell and epithelial/mesenchymal markers in breast cancer cells, depending on culture conditions. Significant differences in marker expression in different growth conditions can be observed, not only between 2D and 3D conditions, but also between different 3D models. Mathematical model of the 3D bio-printed cell culture has been used for analyzing these parameters. The model has been used to compare two different conditions: with a very stiff matrix mimicking alginate hydrogel and a very soft matrix. It substantiates the hypothesis that higher stress requires stronger cell-cell junctions to preserve the integrity of the group of cells. These results may contribute to a better understanding of the differences between various growth models. The manuscript is recommended for publication in Bioengineering. Some comments are suggested below.
(1) In Figure 2B, qPCR analysis demonstrates an increase of both stem cell markers in free floating spheroids. Why does the relative mRNA expression decrease after 2 weeks?
(2) Figure 4 shows the confocal image of a single Z-stack from a 279. Why not reconstruct the 3D model from Z-stack?
(3) The author should indicate the meaning of the significance analysis in the figures, such as *, **, and ****.
(4) In Table 1, for the model parameters assigned to the specific elements, the assumed Young’s modulus of the ECM is set as 56,200 N/m2 for stiff gel and 30 N/m2 for soft gel. Why choose the two values for the model parameters of stress? Please add the corresponding references for supporting.
Author Response
Response to Reviewer 3
In this manuscript, the authors have analyzed the plasticity of expression of stem cell and epithelial/mesenchymal markers in breast cancer cells, depending on culture conditions. Significant differences in marker expression in different growth conditions can be observed, not only between 2D and 3D conditions, but also between different 3D models. Mathematical model of the 3D bio-printed cell culture has been used for analyzing these parameters. The model has been used to compare two different conditions: with a very stiff matrix mimicking alginate hydrogel and a very soft matrix. It substantiates the hypothesis that higher stress requires stronger cell-cell junctions to preserve the integrity of the group of cells. These results may contribute to a better understanding of the differences between various growth models. The manuscript is recommended for publication in Bioengineering. Some comments are suggested below.
(1) In Figure 2B, qPCR analysis demonstrates an increase of both stem cell markers in free floating spheroids. Why does the relative mRNA expression decrease after 2 weeks?
In early stages of sphere development, the expression of stem markers is upregulated, which is associated with the formation of the spheroid. In time, when spheroid grows, the proportion of cells contributing to the spheroid differs. In full grown spheroids (>11 day) the differentiated cells are predominant, while in forming spheroids the stem cells are predominant (10.3390/ijms21155400).
(2) Figure 4 shows the confocal image of a single Z-stack from a 279. Why not reconstruct the 3D model from Z-stack?
Such model was constructed and used for mathematical analysis, however, since it is a live culture, the staining is not very precise, and the informative value of the image is not substantial, it was not presented in the manuscript.
(3) The author should indicate the meaning of the significance analysis in the figures, such as *, **, and ****.
The p-values were provided.
(4) In Table 1, for the model parameters assigned to the specific elements, the assumed Young’s modulus of the ECM is set as 56,200 N/m2 for stiff gel and 30 N/m2 for soft gel. Why choose the two values for the model parameters of stress? Please add the corresponding references for supporting.
The value for the stiff gel was selected to comply with the stiffness of the alginate hydrogel (10.1016/j.mtbio.2024.101071). The value for the soft gel was assigned as the smallest complying to the model parameters.
Round 2
Reviewer 1 Report
Comments and Suggestions for Authors
Revised manuscript can be accepted for publication.
Author Response
We are grateful to the Reviewer for the helpful comments.
Reviewer 2 Report
Comments and Suggestions for Authors
The authors should clearly highlight all revisions in the manuscript using a distinct color and include line numbers for each addressed comment. In its current form, it is difficult to follow the changes made. Additionally, some comments remain unaddressed, such as improving the resolution of Figures 2, 3, and 4. These figures should also be revised and highlighted in color for clarity. Providing a detailed response to each comment, along with visible revisions, will ensure transparency and facilitate the review process.
Author Response
Response 2 to Reviewer 2:
The authors should clearly highlight all revisions in the manuscript using a distinct color and include line numbers for each addressed comment. In its current form, it is difficult to follow the changes made.
All revisions (except for the linguistic, which could be found in docx version with marked all revisions) were marked in yellow. Line numbering is added to each response in the specific answers below.
Additionally, some comments remain unaddressed, such as improving the resolution of Figures 2, 3, and 4. These figures should also be revised and highlighted in color for clarity. Providing a detailed response to each comment, along with visible revisions, will ensure transparency and facilitate the review process.
Since the Reviewer requires addressing the comments citing line numbering, below we repeat our responses from round 1, but including line numbering and additional answer to the comment classified by the Reviewer as unanswered (which was in fact answered, but not addressed, since we had difficulties spotting the problem; however, upon close scrutiny we have identified images of low quality, and replaced them):
Response to Reviewer 2
1. The abstract does not clearly demonstrate how the findings contribute new insights to the field of cancer research, as many studies have already discussed the limitations of 2D and animal models.
The abstract was corrected, highlighting the importance of the study for the proper interpretation of the results obtained using a specific 3D model for cancer research.
Lines: 16, 19-26, 29-33
2. The main objective of the study is not explicitly stated in the abstract, making it challenging to understand the focus of the research.
The abstract was corrected and clarified.
Lines: 16, 19-26, 29-33
3. Coherence and flow are key concepts that need improvement in the abstract.
The abstract was corrected and clarified.
Lines: 16, 19-26, 29-33
4. The introduction lacks proper organization and is notably brief, containing only a limited number of references. Additionally, the identification and explanation of the research gap are insufficiently developed. It would greatly enhance the clarity of the paper if the main objective were articulated more clearly and concisely at the end of the introduction section. Including a more comprehensive review of relevant literature and a more explicit statement of the research gap would provide a stronger foundation for the study.
The introduction was corrected, expanded and clarified, with a more comprehensive review of the literature. The explanation of the research gap was provided.
Lines: 48-59, 63-71, 73-85
5. All abbreviations should be defined upon their first use to enhance clarity and make the content more accessible to a wider audience. (such as: MCF7 (ATCC), T47D (DSMZ) and MDA-MB-231 (ATCC), along with any others used throughout the text.).
The abbreviations were defined were possible, although some are just given names (e.g. cell lines names), not abbreviations.
Lines: 110-11, 131
6. The listing of secondary antibodies includes repeated entries (e.g., Anti-Mouse IgG-AlexaFluor488), which suggests a lack of attention to detail and can confuse readers.
The list was corrected. Lines: 139-140
7. The imaging section lacks detail about the settings or parameters used on the Zeiss Axio observer Z1 LSM 800 confocal microscope, which are essential for reproducibility.
The details were provided. Lines: 185-189
8. The choice of harvesting cells on the 7th day of 3D culture and at ~80% confluency in 2D culture is not explained, which raises questions about the rationale behind these specific time points.
The time points for cell harvest were associated with the spheroid formation process. Early-phase spheroids are defined as spheroids in 5-7 day of culture, and most probably it is also the peak of the expression for stem cell markers - which was confirmed in our preliminary research. Other time points (e.g. 14 days for mature spheroids) were planned according to biological nature of the model, considering the maturation and formation of the cell aggregates. The harvest of the 2D model was set according to standard procedures considering GLP cell culture.
9. The section on cDNA synthesis lacks detail regarding the conditions used (e.g., temperature, time), which are important for understanding the process.
The conditions for the PCR reaction were provided. Lines: 151-154
10. The choice of BIOINK hydrogel is mentioned but not justified. An explanation of why this specific hydrogel was selected would be beneficial.
We selected the BIOINK hydrogel based on alginate due to its well-documented physicochemical properties and widespread application in 3D bioprinting (10.1021/acsabm.3c00984, 10.3390/md21030177, 10.1039/c7bm00765e). Alginate-based hydrogels are known for their mechanical and structural stability, ensuring reproducible experimental results—particularly important in studies involving mathematical modeling. Additionally, the selected hydrogel demonstrates high biocompatibility, creating a cell-friendly microenvironment for long-term cell cultures in 3D structures, as previously documented by our team (Kurzyk A et al 2024 Biofabrication 16 025021) making it a suitable The justification for selecting the BIOINK hydrogel has been added to the revised manuscript in Section 2.7., lines: 170-172
11. The mention of the Abaqus program and GiD program requires more detail about their specific roles in the modeling process and why they were chosen.
It was explained and marked in the text, lines: 236-245
12. The mathematical formulations are missing important references, definition of variables and should be formatted correctly using MathType or an alternative method.
The variables were all defined in the text. The formatting of the equations was done with the current version of MathType. Unfortunately, the most recent versions of MathType do not give very good results. Now, it is done using the embedded in the Word 365 equation editor with latex compatible fonts. We trust it improves the outcome. Of course, Latex is the best, but the requirement is to use Word. We hope it is sufficient for the five equations. We believe that the technical editor will evaluate it and give a good advice.
The changes are not marked in yellow, since there were only formatting changes.
We added one reference concerning the strain increment (linear and nonlinear part), a textbook by Y.C. Fung, Foundations of Solid Mechanics, 2nd Edition, Prentice Hall, 1965.
13. Section 8.8.2: The authors mentioned that the meshing algorithm is sensitive to the shape of the structure but does not explain how this affects the modeling process or results. Add further explanations to figure 1.
We mentioned “it is slightly sensitive”. It does not affect the solution significantly. Providing the mesh is correct, the results are not significantly affected by the mesh density. The mesh density is rather a matter of the resolution of the solution. It is shown in the paper:
Postek E., Sadowski T., Thermomechanical effects during impact testing of WC/Co composite material, COMPOSITE STRUCTURES, DOI: 10.1016/j.compstruct.2020.112054 , Vol.241, pp.112054-1-25, 2020.
In particular, the effects are shown in Figure 15 (mesh definition) and in Table 6 (Mesh dependency check).
It worth to note the numerical solutions are used in qualitative analysis of the system of cells. It is due to inherent simplifications of such a complex system like the living matter.
More explanations concerning meshing are given below (point 14).
14. Has any mesh quality analysis been carried out on the proposed FE model (aspect ratio, Jacobian rate, etc.)? Please justify.
The process of mesh creation starts from building the entire geometry of the system. The geometry is built with GiD preprocessing program (https://www.gidsimulation.com/ ). The program is well tested und used in several industries. The unstructured meshing tool checks the correctness of the elements concerning the basic features like Jacobian and aspect ratio. In the case, the values are incorrect, the mesh is not generated.
The next controlling steps are associated with the program Abaqus. The control of the solution is done at the preprocessing stage and during the finite element system of equations solution phase. During the preprocessing phase, the elements are checked with respect to the mentioned features. In the case of inacceptable parameters, the solution is terminated. It means, the mesh has to be corrected. The failing elements are given in the output from the preprocessing phase.
The resulting FE equations system is checked during the solution phase as well. In particular, it is tested for low Jacobian values. If the solution is not well posed it can be terminated or the number of iterations is increased (an artificial stiffness is added to the system, there are warnings due to the fact).
These issues occurred during the mesh calibration phase. During the final phase, the above mentioned issues did not occur.
LInes: 278-290
15. While it is stated that MCF7 was selected for bioprinting experiments, there is no discussion on why this choice was made in relation to T47D.
The MCF-7 cells line was chosen for 3D printing model because of its excellent spheroid-forming properties. The 3D printing model was done after the spheroid culture analysis. The MCF-7 cell line formed most replicable spheroids and was found to have highest and longer maintaining SOX2 peak during the spheroid culture, which was the reason for selecting this cell line for further experiments (as stated in the manuscript in section 3.1., lines: 306-310).
16. While the results regarding E-cadherin and vimentin expression are stated, there is no detailed interpretation of what these changes imply for the biology of the cell lines studied.
MCF7 and T47D are the standard, commercially available, well characterized, epithelial cell lines with high E-cadherin expression and no vimentin expression, since it is a mesenchymal marker. The rationale for these studies was that one of the goals of our experiments was to observe if there are any signs of EMT, or partial EMT occurring in 3D culture. The EMT and the expression of its markers was described many times and we did not want to repeat this argumentation, although, for clarity, we added some references We did not observe significant changes in vimentin levels, so this is not discussed. However, we feel that changes in E-cadherin expression were discussed extensively in the whole manuscript, because it is the main point of the results.
17. Improve the resolution of Figures 2, 3 and 4.
It is our understanding that the Reviewer's comment pertains to the quality of microscopic images. The resolution of microscopic images in Figures 2 and 3 was improved to the best possible quality, however we want to point out that in some images the signal is very weak and impossible to improve. We cannot change the resolution in Figure 4, it is of its best possible quality, although we want to point out that it represents a thick, live cell culture imaged with objective 10X and as such, it cannot be compared to the quality one may achieve with fixed culture and better magnification.
The Figures 2 and 3 were replaced, but we find it technically hard to mark them in color. We have marked the first line of the Figure legend, to highlight the change.
18. The boundary conditions of the finite element model should be specified more clearly within the model itself.
It has been explained and marked in the text, lines: 255-259
19. The statement that "the maximum stress in cytoplasm is higher in the case of soft ECM" lacks context. Explaining why this difference matters in biological terms would enhance understanding.
The maximal stress was calculated for all cellular elements. While the stress on cortices and cytoskeletal elements is more relevant to the explanation of the differences in E-cadherin expression, the stress on other cellular elements also should be reported, as a part of the comprehensive analysis. In this case, higher stress associated with soft ECM complies with all other results. Additionally, a part of the Discussion section in which the effect of mechanical stress on cell junctions and cytoskeleton is commented has been expanded in a revised version of the manuscript.
Lines: 537-539
20. It is essential to include a conclusion at the end of the manuscript.
A section ‘Conclusions’ has been included.
21. The manuscripts are excessively lengthy, containing numerous redundancies and failing to maintain a clear focus throughout. This can make it challenging for readers to grasp the main points and key findings effectively. Streamlining the content and eliminating repetitive information would enhance clarity and engagement.
In the corrected version we tried to eliminate redundancies and improve clarity.
Round 3
Reviewer 2 Report
Comments and Suggestions for Authors
The authors have effectively addressed the majority of my concerns.